# FSTL1 Suppresses Triple-Negative Breast Cancer Lung Metastasis by Inhibiting M2-like Tumor-Associated Macrophage Recruitment toward the Lungs

**DOI:** 10.3390/diagnostics13101724

**Published:** 2023-05-12

**Authors:** Ying Yang, Tao Lu, Xiaowei Jia, Yan Gao

**Affiliations:** 1Beijing Key Laboratory of Cancer Invasion and Metastasis Research, Department of Human Anatomy, School of Basic Medical Sciences, Capital Medical University, Beijing 100069, China; 2Department of Human Anatomy, Capital Medical University, No. 10 Xitoutiao, You’anmenwai, Fengtai District, Beijing 100069, China

**Keywords:** triple-negative breast cancer, breast cancer lung metastasis, follistatin-like 1 (FSTL1), tumor microenvironment, M2-like tumor-associated macrophages

## Abstract

Immune cell infiltration into the tumor microenvironment is associated with cancer prognosis. Tumor-associated macrophages play essential roles in tumor initiation, progression, and metastasis. Follistatin-like protein 1 (FSTL1), a widely expressed glycoprotein in human and mouse tissues, is a tumor suppressor in various cancers and a regulator of macrophage polarization. However, the mechanism by which FSTL1 affects crosstalk between breast cancer cells and macrophages remains unclear. By analyzing public data, we found that FSTL1 expression was significantly low in breast cancer tissues compared to normal breast tissues, and high expression of FSTL1 in patients indicated prolonged survival. Using flow cytometry, we found that total and M2-like macrophages dramatically increased in the metastatic lung tissues during breast cancer lung metastasis in *Fstl1*^+/−^ mice. Transwell assay in vitro and q-PCR experimental results showed that FSTL1 inhibited macrophage migration toward 4T1 cells by decreasing CSF1, VEGF-α, and TGF-β secretion in 4T1 cells. We demonstrated that FSTL1 inhibited M2-like tumor-associated macrophage recruitment toward the lungs by suppressing CSF1, VEGF-α, and TGF-β secretion in 4T1 cells. Therefore, we identified a potential therapeutic strategy for triple-negative breast cancer.

## 1. Introduction

In 2016, the incidence rate of breast cancer in China reached 29.05/100,000, and the mortality rate reached 6.39/100,000, becoming the fifth most common cancer in China and the most common cancer in women [1]. Breast cancer can be classified into several molecular subtypes based on histological features, including hormone receptor-positive, human epidermal growth factor receptor-2 overexpressing (HER2+), and triple-negative breast cancer (TNBC). TNBC, which barely expresses ER, PR, or HER2, is the most malignant and intractable type of breast cancer. It is associated with a poor prognosis due to metastasis in the early stage [2,3,4]. Therefore, the suppression of TNBC metastasis is an important issue.

Breast cancer development, metastatic ability, and therapeutic reactivity rely on both cancer cells and their interactions with the surrounding microenvironment [5]. The tumor microenvironment (TME) comprises cellular components (such as cancer cells, immune cells, fibroblasts, and endothelial cells), growth factors, proteases, and extracellular matrix (ECM) [6,7,8]. Innate immune cells are highly represented in the TME, and the most abundant are tumor-associated macrophages (TAMs). TAMs, a subgroup of macrophages, can conventionally be subdivided into M1-like and M2-like TAMs (Appendix A). Activated M1-like TAMs exert antitumor effects [9]. In contrast, M2-like TAMs undergo a protumor process with multiple functions in tumorigenesis and metastasis [10]. M2-like TAMs, activated by IL-4, IL-13, IL-10, and M-CSF/CSF-1, can secrete cytokines such as TGF-β, VEGF, and EGF, which would further promote tumor pre-metastatic niche formation and angiogenesis around the niche [11].

Follistatin-like protein 1 (FSTL1) is a widely expressed glycoprotein in human and mouse tissues with various functions (Appendix A). It is essential for multiple processes, including cellular biology, organ development, carcinogenesis, and metastasis [12]. FSTL1, a well-known tumor suppressor in different types of cancer [13,14,15], has recently been reported as a potential regulator of macrophage polarization [16]. We have previously demonstrated that FSTL1 deficiency accelerates the growth of breast cancer lung metastatic tumors, but not primary tumor growth [17]. However, the dynamic and complex changes in the TME of lung tissues require clarification.

In this study, we analyzed the potential relationship between FSTL1 and the lungs’ TME in breast cancer metastasis. The results showed that FSTL1 reduced the population of M2-like TAMs in the lung and inhibited the secretion of CSF1, VEGF-α, and TGF-β cytokines in 4T1 TNBC cells. Immune cell infiltration in the TME is closely related to tumor prognosis [18], and FSTL1 may be able to evolve cancer immunotherapy to treat malignant tumors and their metastasis.

## 2. Materials and Methods

### 2.1. Reagents and Facilities

RPMI-1640 medium was purchased from Thermo Fisher Scientific (Waltham, MA, USA). Reagent-extracting RNA was purchased from Sangon Biotech (Shanghai, China). The following cytometry-labeled antibodies were used: PE anti-mouse F4/80 Antibody (Catalog: #123110; BioLegend, San Diego, CA, USA), FITC anti-mouse CD206 (MMR) antibody (Catalog: #141704; BioLegend), and Fixable Viability Stain 450 (FVS450, Catalog: #562247; BD Biosciences, Franklin Lakes, NJ, USA). Recombinant mouse IL-4 (Catalog: #214-14; PeproTech, Cranbury, NJ, USA) and recombinant mouse FSTL1 proteins (Catalog: #1738-FN-050; R&D Systems, Minneapolis, MN, USA).

### 2.2. Experimental Animals

129 Background *Fstl1*^flox/+^ mice were backcrossed with BALB/c background mice for more than 10 generations, and the experimental animals were obtained: BALB/c background *Fstl1*^+/−^ mice. Mice were purchased from the Model Animal Research Center of Nanjing University and bred at our animal institution under controlled conditions with a 12 h/12 h light–dark cycle. All animal experiments were performed in accordance with the Administration Regulations on Laboratory Animals of Beijing Municipality. All the animals were raised under specific pathogen-free (SPF) conditions and provided with adequate irradiated food and water. All animals used were healthy female BALB/c mice (6–8 weeks).

### 2.3. Experimental Cell Lines

The 4T1 TNBC cell line was a kind gift from Professor Wen Ning of Nankai University (Tianjin, China). Cells were grown in RPMI 1640 containing 10% fetal bovine serum (FBS) and 1% penicillin/streptomycin (P/S) and were cultured at 37 °C in 5% CO_2_. The Macrophage RAW264.7 cell line was a gift from Professor Ying Sun, and the Ana-1 cell line was a gift from Professor De-Shan Zhou of Capital Medical University (Beijing, China). These two macrophage cell lines were cultured in a sterile RPMI 1640 medium containing 10% FBS and incubated at 37 °C in 5% CO_2_. Cells were cultured for more than two generations and digested for subsequent experiments.

### 2.4. Mouse Genotyping

Mouse tail (0.5–1 cm) was cut and digested for genomic DNA solution. DNA solution (50 μL) was removed from the preceding solution, double the volume of 100% alcohol was added, and the extracted DNA was visible to the naked eye. The samples were washed with 75% alcohol, and DNA was diluted to a certain concentration for further testing. FSTL1 primers: Forward: 5′-CTCCCACCTTCGCCTCTAAC-3′; Reverse: 5′-CGGCTAGGAAAGACTTGGAA-3′.

### 2.5. Breast Cancer Model Mice

Six-to-eight-week-old healthy female BALB/c background wild-type mice (WT; control group) and *Fstl1*^+/−^ mice (experimental group) were age- and weight-matched to build breast cancer models. On the 0th day, the mice were anesthetized with 0.25 mg/kg pentobarbital sodium and injected with 1 × 10^6^ 4T1 cells into the second fat pad under a stereoscope. Mice were housed and fed in individual ventilation cages (IVC). When the tumors could be touched in situ, the length and width of the tumors were measured using calipers every three days. Tumor volume was calculated using the following formula: tumor volume = length × width^2^/2. On the 28th day, the mice were euthanized. Tumors in situ were excised, tumor weights were measured, and the lungs were removed from the thoracic cavities of the mice to prepare for the total mRNA, protein assay, and other experiments.

### 2.6. Hematoxylin–Eosin Staining

Fixed lung tissues were embedded in paraffin and prepared as slices. The lung tissue slices were placed in distilled water and stained with a hematoxylin solution for several minutes. They were then placed in acidic and ammonia water for color separation for a few seconds each. This was followed by rinsing with water for 1 h and then with distilled water. The slices were dehydrated in 70% and 90% alcohol for 10 min each and stained with an alcohol eosin staining solution for 2–3 min. Stained sections were dehydrated using pure alcohol and made transparent using xylene. Finally, a drop of gum was placed on the sections, and a cover glass was placed to seal them.

### 2.7. Quantitative Real-Time Polymerase Chain Reaction

The following primers were used in q-PCR. β-actin, Forward: 5′-CATCCGTAAAGACCTCTATGCCAAC-3′; Reverse: 5′-ATGGAGCCACCGATCCACA-3′; IL-4, Forward: 5′-GGTCTCAACCCCCAGCTAGT-3′; Reverse: 5′-GCCGATGATCTCTCTCAAGTGAT-3′; IL-13, Forward: 5′-CCTGGCTCTTGCTTGCCTT-3′; Reverse: 5′-GGTCTTGTGTGATGTTGCTCA-3′; Arg-1, Forward: 5′-CAAGGTGATGGAAGAGACCTT-3′; Reverse: 5′-TAAGGTAGTCAGTCCCTGGCTT-3′; IL-10, Forward: 5′-ATGCTGCCTGCTCTTACTGACTG-3′; Reverse: 5′-CCCAAGTAACCCTTAAAGTCCTGC-3′; CCL2, Forward: 5′-TTAAAAACCTGGATCGGAACCAA-3′; Reverse: 5′-GCATTAGCTTCAGATTTACGGGT-3′; CCL5, Forward: 5′-GCTGCTTTGCCTACCTCTCC-3′; Reverse: 5′-TCGAGTGACAAACACGACTGC-3′; CSF1, Forward: 5′-GTGTCAGAACACTGTAGCCAC-3′; Reverse: 5′-TCAAAGGCAATCTGGCATGAAG-3′; VEGF-α, Forward: 5′-GCACATAGAGAGAATGAGCTTCC-3′; Reverse: 5′-CTCCGCTCTGAACAAGGCT-3′; and TGF-β, Forward: 5′-CTCCCGTGGCTTCTAGTGC-3′; Reverse: 5′-GCCTTAGTTTGGACAGGATCTG-3′. The q-PCR analysis was performed as described previously [19].

### 2.8. Western Blot Analysis

Lung and breast tissues were lysed in RIPA buffer (Applygen, Beijing, China) containing a proportional mixture of a protease inhibitor (1:50) and a protein phosphatase inhibitor (1:100). Mouse FSTL1 antibody (Catalog: AF1738; R&D Systems, Inc., Spokane Valley, Washington, DC, USA; dilution 1:1000), β-actin (Catalog: #4970; Cell Signaling Technology Corp., Danvers, MA, USA; dilution 1:2000), Arg-1 (Catalog: sc-271430; Santa Cruz Biotechnology, Santa Cruz, CA, USA; dilution 1:1000), MMP9 (Catalog: ab58803; Abcam Technology Group Inc., Cambridge, UK; dilution 1:1000), and TGFβ (Catalog: ab66043; Abcam Technology Group Inc., Cambridge, UK; dilution 1:1000). Western blotting was performed using standard protocols as previously described [19].

### 2.9. Flow Cytometry

Cytometrically analyzed single-cell suspensions were prepared from lung tissues. Flow cytometry was conducted to identify immune cell populations. First, cells were incubated with fixable viability strain 450 (FVS450) for 15 min. The cells were then incubated with a monoclonal antibody against CD16/32 for 10 min to block the cellular Fc receptor. Finally, the cells were stained with PE anti-mouse F4/80 and FITC anti-mouse CD206 (MMR) antibodies for 60 min. CD206 is a transmembrane protein that requires both a cellular surface and intracellular staining. If necessary, the samples were incubated overnight with 2% paraformaldehyde. Data were integrated using the Tree Star FlowJo software.

### 2.10. Macrophage Migration Assay

To mimic the lung tumor microenvironment, the 8.0 μm Transwell was used to study the co-culture of 4T1 TNBC cells and RAW264.7/Ana-1 macrophages. RAW264.7/Ana-1 cells (3 × 10^4^ cells/well, 150 μL 1640) were placed in the upper well, whereas 4T1 cells (1.2 × 10^5^ cells/well, 500 μL complete medium containing 10% FBS) were placed in the lower well. RAW264.7/Ana-1 macrophages:4T1 cells (1:4). FSTL1 and IL4 were separately or simultaneously administered into the lower well. After 12 h co-culture, the 8.0 μm Transwell was washed and stained with 4% paraformaldehyde containing 0.1% crystal violet for 30 min. Five fields per slide were observed using the 10× objective lens of a microscope, and the number of migrated macrophages was calculated using Photoshop software and compared with that of the blank control.

### 2.11. Cell Proliferation Assay

4T1 breast cancer cells and RAW264.7 macrophages were grown and digested with 0.25% trypsin for cell proliferation analysis. 4T1 (1.2 × 10^5^ cells/well) and RAW264.7 cells (3 × 10^4^ cells/well) were inoculated in 96-well plates and were cultured for 12 h at 37 °C in 5% CO_2_. Finally, the absorption data of the CCK8 values were assayed using a Thermo Scientific Multiskan MK3 Enzyme Mark Instrument.

### 2.12. Database

UALCAN is used for digging TCGA (The Cancer Genome Atlas) and CPTAC (Clinical Proteomic Tumor Analysis Consortium) data. Available from: http://ualcan.path.uab.edu (accessed on 10 March 2020). Kaplan–Meier survival analysis was used to assess the survival rates of 21 cancer types, including breast cancer. Website: https://kmplot.com/analysis/ (accessed on 10 March 2020).

### 2.13. Statistical Analysis

Statistical analysis was performed using Microsoft Excel 2013 (Microsoft Corp., Redmond, Washington, DC, USA), GraphPad Prism6.0 (GraphPad Software Inc., San Diego, CA, USA), and ImageJ. All results were calculated as mean ± standard deviation, and all analyses were considered statistically significant at *p* < 0.05. All graphs show mean ± SD.

## 3. Results

### 3.1. FSTL1 mRNA Expression Decreases in Human Breast Cancer and Its Various Subtypes

Our laboratory previously reported that FSTL1 deficiency accelerates the growth of breast cancer cells at metastatic lung sites. To confirm the suppressive role of FSTL1 in human breast cancer, we collected invasive breast cancer samples from TCGA and sorted them into different types. The results showed that FSTL1 expression was significantly lower in all invasive breast cancers compared to that in normal breast tissues (Figure 1A). All individual stages of invasive breast cancer lacked FSTL1 expression compared with normal breast tissues (Figure 1B). The same result was observed for different nodal metastatic statuses (Figure 1C). Furthermore, invasive breast cancer tissues sorted by luminal, HER2, and TNBC tissues showed lower FSTL1 expression than normal breast tissues (Figure 1D). Considering that FSTL1 is deficient in all types of human breast cancers, we regard FSTL1 as a possible human breast cancer suppressor.

### 3.2. Patients with Breast Cancer and High FSTL1 Expression Showed Prolonged Survival

To further predict the role of FSTL1 in breast cancer, we analyzed the survival rates of patients with breast cancer. As shown in Figure 1E, although patients with breast invasive carcinoma (BRCA) from the TCGA database with high and low/medium FSTL1 expression had the same survival rate before 4000 days, they seemed to have a better survival rate over a more extended period. Furthermore, fewer patients with TNBC had high FSTL1 expression and better survival (Figure 1F). Another survival curve determined by the Kaplan–Meier plotter showed that the overall survival (OS) rate decreased dramatically in patients with lower FSTL1 expression compared to those in the higher expression groups (Figure 1G). The same trend was observed in patients with breast cancer positive metastases (Figure 1H). Therefore, FSTL1 could be potentially beneficial for the prognosis of patients with breast cancer, especially for TNBC patients.

### 3.3. FSTL1 Does Not Affect the Proliferation of TNBC In Situ, but Remarkably Increases Its Lung Metastasis

To further investigate the functional mechanism of FSTL1 in breast cancer, we established a breast cancer lung metastasis mouse model as described previously. WT mice served as a control group, and *Fstl1*^+/−^ mice (as *Fstl1*^+/−^ mice cannot survive after birth) served as an experimental group, which exhibited low FSTL1 expression. The confirmed animal results were the same as Zhang et al., 2018 [17]. We then tested the proliferation of 4T1 cells treated with recombinant mouse FSTL1 protein, and the results revealed that FSTL1 did not affect the proliferation of 4T1 cells (Figure 2A). We also analyzed the expression of several Epithelial–Mesenchymal Transition (EMT)-related markers, including N-cadherin and E-cadherin, in 4T1 cells treated with different FSTL1 concentrations and found no significant difference between the control and FSTL1-treated groups (Figure 2B). The evidence above indicates that FSTL1 does not affectcancer proliferation and migration abilities, but increases metastasis to the lungs.

### 3.4. Fstl1^+/−^ Mice Exhibit Increased M2 Macrophages Deposition in the Lungs

Macrophages are a large population of immune cells that are important components of the tumor microenvironment. As we hypothesized that FSTL1 plays a vital role in the TME, we next investigated the macrophage ratios in the lungs on day 0 and day 28. As expected, the results displayed that total and M2 macrophage rates did not differ between the two groups on day 0 (Figure 3A,B), but significantly increased on day 28 (Figure 3A,B). Furthermore, the expression of the M2 macrophage-related markers IL4, IL13, Arg-1, and IL10 also increased remarkably on day 28 (Figure 3C–E). Expression of metastasis-related markers TGFβ and MMP9 (Figure 3F,G) was also significantly upregulated in the lungs of *Fstl1*^+/−^ mice to support the aforementioned results. The above data show that FSTL1 promoted the accumulation of macrophages, specifically tumor-promoting M2 macrophages, in the TME of the lungs.

### 3.5. FSTL1 Inhibited M2-like TAMs Migration toward 4T1 TNBC Cells

To mimic macrophage recruitment in vivo and the crosstalk between immune and cancer cells, a co-culture Transwell assay was designed with macrophage cell lines in the upper well and 4T1 cells in the lower well (Figure 4B,E). The migration ability of macrophages did not differ among the control, FSTL1, IL4, and IL4 + FSTL1 groups (Figure 4A,C,D,F). The number of migrating cells significantly increased in the 4T1 group but decreased dramatically in the 4T1 + FSTL1 group (Figure 4A,C,D,F). These results indicate that 4T1 cells are attractive factors for macrophages and that FSTL1 could suppress this phenomenon. Next, the number of migrating cells increased in the 4T1 + IL4 groups and decreased in the 4T1 + IL4 + FSTL1 group (Figure 4A,C,D,F). As IL4 is an activator of M2-like TAMs, these results further demonstrate that FSTL1 is a suppressor of M2-like TAMs.

### 3.6. FSTL1 Inhibited the Secretion of CSF1, VEGF-α, and TGF-β in 4T1 TNBC Cells

These results revealed that cytokine secretion by 4T1 cells played a vital role after treatment with recombinant mouse FSTL1 protein. To investigate how 4T1 cells affect M2-like TAMs and the role of FSTL1 during breast cancer metastasis, we treated 4T1 cells with FSTL1/IL4 and detected the expression of some M2-like TAMs-associated factors using q-PCR (Figure 5A). The CCL2 and CCL5 mRNA expression increased significantly with IL4 alone, and FSTL1 did not suppress this process (Figure 5B,C). The CSF1, VEGF-α, and TGF-β mRNA expression also increased dramatically, but FSTL1 could suppress their secretion, which was different with CCL2 and CCL5 (Figure 5D–F). Thus, we demonstrated that FSTL1 could inhibit the secretion of CSF1, VEGF-α, and TGF-β cytokines produced by 4T1 TNBC cells, which further inactivated M2-like TAMs (Figure 5G).

## 4. Discussion

This study is a sequential extension of our previous study. Our previous data showed that deficiency in FSTL1 could accelerate the growth of breast cancer lung metastatic tumors, but not primary tumor growth [17]. However, the underlying mechanisms remain unclear. In our study, 4T1 breast cancer cells did not express FSTL1, and their proliferation and EMT markers did not change after treatment with recombinant FSTL1. Thus, we speculated that FSTL1 might affect the proliferation of metastatic cancer cells by altering the lung TME, which might be a critical cause of this phenotype. In the present study (Appendix A), we analyzed the potential function of FSTL1 in patients with breast cancer using a public database. These results indicate the potential benefit of FSTL1 in patients with breast cancer, especially in those with TNBC. Next, we confirmed that FSTL1 did not affect the proliferation of 4T1 cells in vivo or in vitro. However, metastatic cancer in the lungs increased remarkably in the FSTL1-deficient mice. Thus, we hypothesized that FSTL1 is related to TME in the lungs.

The tumor microenvironment (TME) remains poorly understood because of its complex components, especially its immune status. Tumor-associated macrophages (TAMs) are the predominant elements [20,21,22]. TAMs are essential drivers of tumor progression, metastasis, and resistance to therapy. Antitumor M1-like and protumor M2-like TAMs coexist in the TME [23]. Therefore, the percentage of M2-like TAMs in the TME directly affects cancer metastasis. Our results showed that M2-like TAMs were significantly increased in the FSTL1-deficient lungs during breast cancer transfer. Furthermore, IL4 and IL13, which stimulate M2-like TAMs [24,25], were also increased in metastatic lung tissues. Accordingly, Arg1, IL10, MMP-9, and TGF-β, the stimulators of M2-like TAMs [11,26,27], were dramatically upregulated in metastatic lung tissues. These results demonstrated that FSTL1 is a suppressor of M2-like TAMs in the lung TME. To confirm this result, we designed a co-culture Transwell assay. IL4 served as stimuli for macrophages and 4T1 cells, whereas 4T1 cells served as an attractive factor. The results showed that FSTL1 suppressed the macrophages attracted to 4T1 and IL4-treated 4T1 cells.

In the typical course of TAMs polarization, CCL2 and CCL5 chemokines recruit monocytes from the circulatory system to the organs [28]. The monocytes that infiltrate the organ are polarized into M2-like TAMs after stimulation by some cytokines (such as IL4, IL10, CSF1, VEGF-α, and TGF-β) [29,30,31,32]. Our study found that 4T1 cells automatically secreted CCL2, CCL5, CSF1, VEGF-α, and TGF-β, which significantly increased after treatment with IL4. Furthermore, FSTL1 inhibited the increase in CSF1, VEGF-α, and TGF-β induced by IL4, except in CCL2 and CCL5. Thus, our research drew a relatively well-developed pattern to elucidate the mechanism by which FSTL1 inhibited the crosstalk between TNBC cells and M2-like TAMs. We observed a remarkable increase in metastatic TNBC from the breast to the lung tissue in the FSTL1-deficient mice. The TNBC cell line, 4T1, can secrete CCL2, CCL5, CSF1, VEGF-α, and TGF-β, which are the stimuli of M2-like TAMs. CCL2 and CCL5 recruited monocytes from the circulatory system into the lungs. Thereafter, CSF1, VEGF-α, and TGF-β promoted monocyte polarization into M2-like TAMs, which would secrete Arg1 and IL10 to further promote TNBC progress by establishing an immunosuppressive TME. Next, CCL2, CCL5, CSF1, VEGF-α, and TGF-β cytokines were released by TNBC. Therefore, a loop was formed between the TNBC and M2-like TAMs. FSTL1 can inhibit the secretion of CSF1, VEGF-α, and TGF-β in TNBC to interrupt the process and sequentially suppress the lung metastasis of TNBC. Thus, the present study suggests that FSTL1 is a potential marker for predicting prognosis and a potential treatment agent for inhibiting lung metastasis in breast cancer.

Triple-negative breast cancer (TNBC) is characterized by aggressive biology and a high risk of distant recurrence [33,34,35]. Immune checkpoint inhibitors such as anti-programmed cell death 1 (PD-1) and anti-PD-ligand 1 (PD-L1) agents are currently being investigated for the treatment of TNBC. Tumor-associated macrophages (TAMs) directly and indirectly modulate PD-1 and PD-L1 expression in the cancer microenvironment [36]. Our laboratory has previously reported that FSTL1 deficiency could impair T cell development in the thymuses and decrease T cell ratios in the lungs [37], and has now discovered that FSTL1 can also reduce M2-like TAM ratios at the metastatic lung sites. As reported, M2–TAM subsets could be redistributed by lactic acid levels to upregulate PD-L1 and assist tumor immune escape [38]. In the future, FSTL1 may arise as a cellular or molecular TAMs agonist or antagonist in immune checkpoint therapy of breast cancer.

## Figures and Tables

**Figure 1 diagnostics-13-01724-f001:**
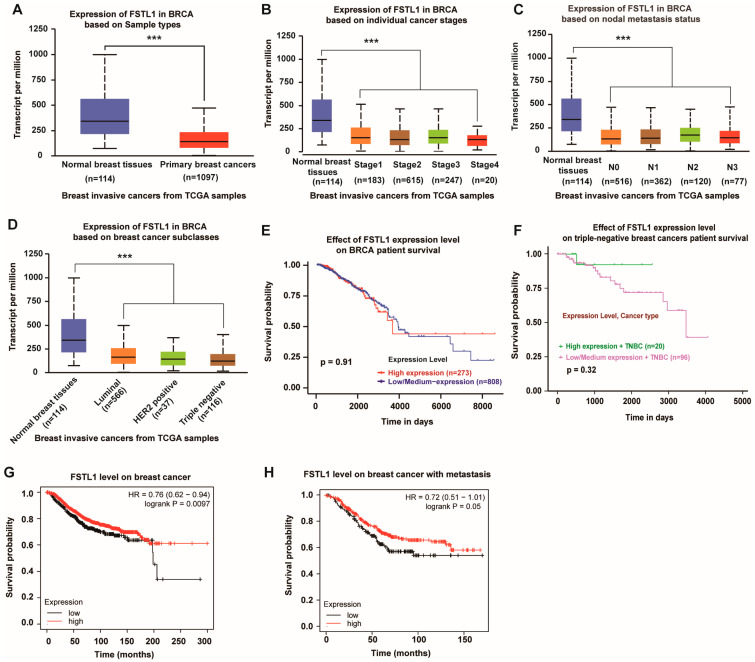
Lower FSTL1 expression in breast cancer tissues compared to normal breast tissues, and high expression of FSTL1 meant prolonged survival. (**A**–**D**) FSTL1 mRNA expression decreased in primary breast cancer. (**E**,**F**) FSTL1 had no significant effect on the survival rate of patients with breast invasive carcinoma (BRCA) and TNBC. (**G**,**H**) High FSTL1 expression could increase the survival rate of patients with breast cancer and those with positive nodal metastasis. *** *p* < 0.001.

**Figure 2 diagnostics-13-01724-f002:**
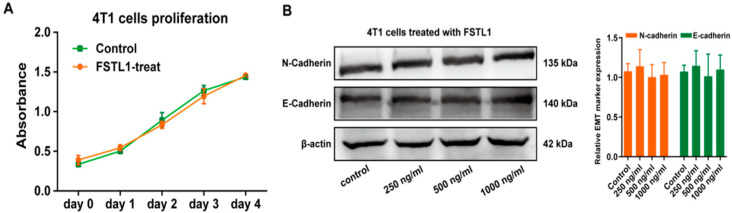
FSTL1 had no effect on the proliferation and EMT markers of 4T1 cells. (**A**) The proliferation ability of 4T1 TNBC cells showed no change after 600 ng/mL FSTL1 treatment for 1–4 d. (**B**) Under different FSTL1 concentration treatments, expression of 4T1 EMT markers, including E-cadherin and N-cadherin, showed no change after 12 h treatment.

**Figure 3 diagnostics-13-01724-f003:**
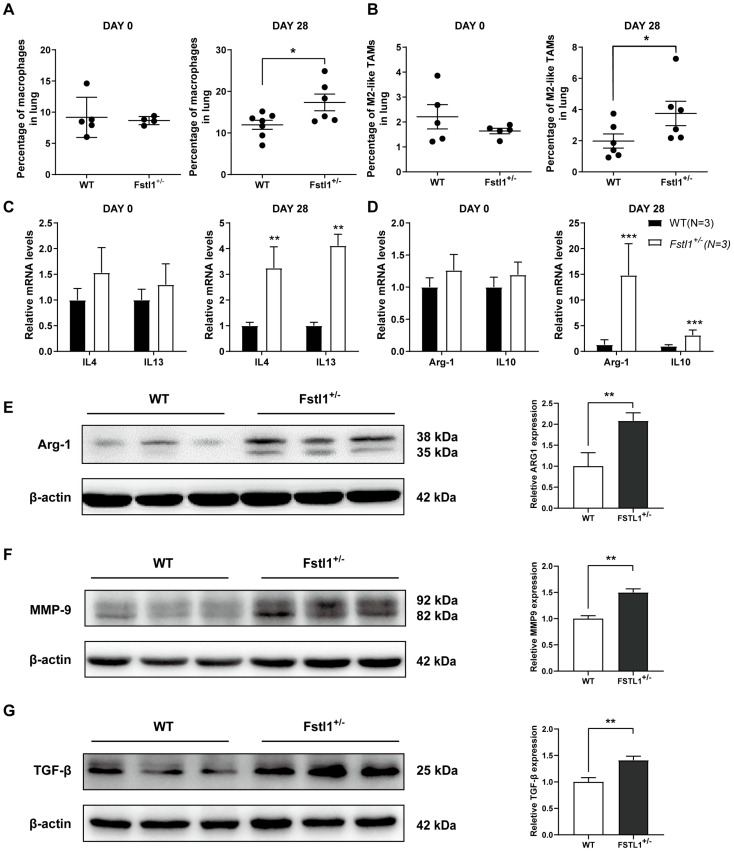
Total and M2 macrophage ratios increased in breast cancer lung metastasis in *Fstl1*^+/−^ mice. (**A**,**B**) Total (F4/80+) and M2 (F4/80+CD11c-CD206+) macrophage ratios in WT and *Fstl1*^+/−^ mice on the 0th and 28th day. (**C**–**E**) Expression of several M2-related macrophage markers (IL4, IL13, Arg-1, and IL10) in WT and *Fstl1*^+/−^ mice on the 0th and 28th day. (**F**,**G**) Western blot of metastasis-related markers (TGFβ and MMP9) of WT and *Fstl1*^+/−^ mice on the 28th day. * *p* < 0.05, ** *p* < 0.01, *** *p* < 0.001.

**Figure 4 diagnostics-13-01724-f004:**
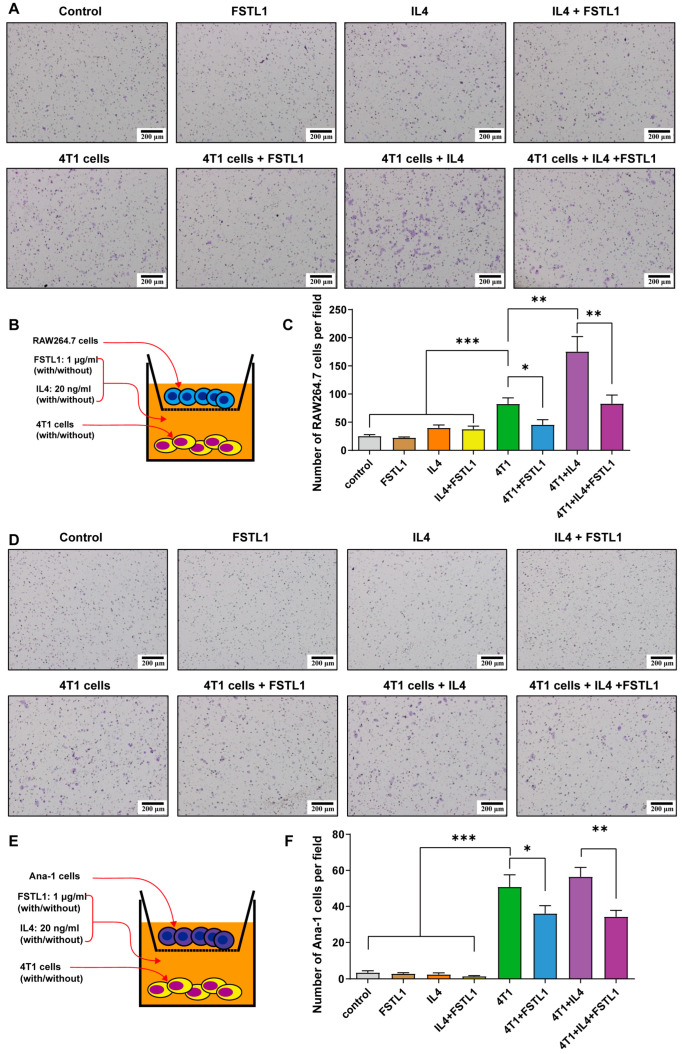
FSTL1 inhibited macrophage migration toward 4T1 breast cancer cells. (**A**,**B**,**D**,**E**) RAW264.7/Ana-1 (the upper transwell) and 4T1 cells (the lower transwell), stimulator: FSTL1 (1 μg/mL) ± IL4 (20 ng/mL), transmembrane macrophage morphology was observed, and numbers (5 fields per slide) were calculated after 12 h co-culture. (**C**,**F**) Migrated RAW264.7/Ana-1 macrophages toward 4T1 breast cancer cells decreased in FSTL1 (1 μg/mL) treatment group. * *p* < 0.05, ** *p* < 0.01, *** *p* < 0.001.

**Figure 5 diagnostics-13-01724-f005:**
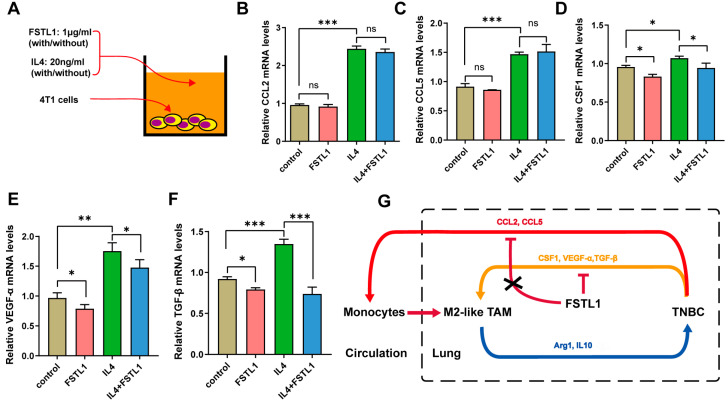
Recombinant mouse FSTL1 protein inhibited several cytokines produced by 4T1 TNBC cells. (**A**) FSTL1 (1 μg/mL)-treated 4T1 cells with or without IL4 (20 ng/mL) in vitro. Cytokine secretions: (**B**) CCL2, (**C**) CCL5, (**D**) CSF1, (**E**) VEGF-α, and (**F**) TGF-β. (**G**) Schematic representation of the effect of FSTL1 on breast cancer metastatic tumor growth. The model depicts the suppression of CSF1, VEGF-α, and TGF-β secretion in 4T1 breast cancer cells by FSTL1, which decreases macrophage recruitment toward TNBC cells. * *p* < 0.05, ** *p* < 0.01, *** *p* < 0.001.

## Data Availability

The data presented in this study are available in the present manuscript and in the Appendix A.

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
