# Peer review of "FSTL1 Suppresses Triple-Negative Breast Cancer Lung Metastasis by Inhibiting M2-like Tumor-Associated Macrophage Recruitment toward the Lungs"

_diagnostics, 2023, doi:10.3390/diagnostics13101724_

Round 1

Reviewer 1 Report

I believe the manuscript has some flaws that need to be addressed before it can published:

1.       Even though English is not my first language and I cannot judge it, I feel the manuscript would benefit from the proofreading for typos, scientific and grammar mistakes which there are multiple, few examples are: line 293:” had been next discovered”; line 85 “DNA was prepared to be diluted”, line 33: I think “newly diagnosed” shouldn’t have a dash, etc.

2.       Line 141: what is enzyme-labeled instrument? It is not clear which assay was used for cell proliferation.

3.       lines 52-53, citation 17: it has been already established that “deficiency of FSTL1 could accelerate the growth of 52 breast cancer lung metastatic tumor”; Why the same experiment with a similar outcome is repeated in this manuscript (Fig 2)? Fig.2: please quantify data from panel D if it’s any different from previously shown in [17].

4.       Fig.1G has already been published as Fig.1E in [17].

5.       Fonts size on Fig.1 needs to be increased.

6.       I understand that FSTL1 is shown to be a tumor suppressor in this study but it is unclear what outcome authors propose by (line 295-296) “FSTL1 immune checkpoint value in breast cancer therapy as TAMs agonist or antagonist cellularly or molecularly”. It is not clear, do authors consider FSTL1 as a marker for therapy, or a prognostic marker, or a treatment agent, or a treatment target?

Reviewer 2 Report

The manuscript by Gao et al., is well-scheduled and performed. The authors convincingly show that FSTL1 attenuates the metastasis of TNBC cells to the lung and suggest a plausible mechanism. Due to the difficulties in treating TNBC, new developments in therapy options are urgently needed.

Comments

The data is convincing and experimental conditions adequately described. Since the ECM component of the TME is an important component of the suggested mechanisms it should be further introduced and more extensively disccussed.

The English needs editing, especially regarding the style as at points it makes the text difficult to understand.

e.g title in line 178 and throuoghout

-line 184 "more rough" please explain

-line 185, "more significant", here and throughout, should be changed to "with a higher significance." (something is significant or not)

-In all contexts edit that FSTLS1 decreases the ability of TNBC cells to metastasize

- title for 3.4 is another example of the awkward style. Please change to "FSTK1+/- mice exhibit increased macrophage deposition to the lungs"

Reviewer 3 Report

Yang et al. report an interesting original research about the roles of Follistatin-like protein 1 in triple negative breast carcinoma metastases to the lung. 

I think that the work is suitable for publication in Diagnostics journal but it requires some minor changes: 

-       Introduction: please do a figure showing the different types of tumor-associated macrophages and their phatophysiology

-       Introduction: please do a figure showing the different roles in pathophysiology of the Follistating-like protein 1

-       Discussion: please do a table listing all the main results of the study

Round 2

Reviewer 1 Report

I am still convinced that the following has to be addressed: two of the observations in this paper have been published before by the authors:

1) experiment described in Fig.2 A - E was performed exactly the same way in Zhang et al., 2018; I am not sure it needs to be repeated exactly the same way; the EMT markers done with cell culture is the new data indeed, and I have no objections to that.

2) I was probably not clear; I meant that Figure 1G was already published as Fig 1E in Zhang et al., 2018, in exactly same way, with nothing changed. I did not mean Fig. 1E from the current manuscript.

Authors state that the grammar of the whole document was corrected but I still see things like "0th day" which in my opinion need to be corrected. It's better if professional scientific proofreading of the whole manuscript is done.

Round 3

Reviewer 1 Report

The authors addressed majority of the comments.